# Polyphasic OKJIP Chlorophyll *a* Fluorescence Transient in a Landrace and a Commercial Cultivar of Sweet Pepper (*Capsicum annuum*, L.) under Long-Term Salt Stress

**DOI:** 10.3390/plants10050887

**Published:** 2021-04-28

**Authors:** Pasquale Giorio, Mohamed Houssemeddine Sellami

**Affiliations:** National Research Council of Italy, Institute for Mediterranean Agricultural and Forestry Systems (CNR-ISAFOM), Piazzale Enrico Fermi, 1 Località Porto del Granatello, 80055 Portici, NA, Italy; hsellemi@yahoo.fr

**Keywords:** JIP test, Kautsky effect, photosynthesis, PSII photosystem, salinity, stress tolerance

## Abstract

In a soilless long-term salt-stress experiment, we tested the differences between the commercial sweet pepper cultivar “Quadrato d’Asti” and the landrace “Cazzone Giallo” in the structure and function of PSII through the JIP test analysis of the fast chlorophyll fluorescence transients (OKJIP). Salt stress inactivated the oxygen-evolving complex. Performance index detected the stress earlier than the maximum quantum yield of PSII, which remarkably decreased in the long term. The detrimental effects of salinity on the oxygen evolving-complex, the trapping of light energy in PSII, and delivering in the electron transport chain occurred earlier and more in the landrace than the cultivar. Performance indexes decreased earlier than the maximum quantum yield of PSII. Stress-induced inactivation of PSII reaction centers reached 22% in the cultivar and 45% in the landrace. The resulted heat dissipation had the trade-off of a correspondent reduced energy flow per sample leaf area, thus an impaired potential carbon fixation. These results corroborate the reported higher tolerance to salt stress of the commercial cultivar than the landrace in terms of yield. PSII was more affected than PSI, which functionality recovered in the late of trial, especially in the cultivar, possibly due to heat dissipation mechanisms. This study gives valuable information for breeding programs aiming to improve tolerance in salt stress sensitive sweet pepper genotypes.

## 1. Introduction

The world population will exceed 9 billion in 2050 [1] when 50% of croplands may not produce food because of soil salinization [2]. Local ecotypes or landraces have stable though moderate yield capacity and potentially high-stress tolerance [3]. Thus, they represent valuable genetic reservoirs of genes underlying stress tolerance. Uncovering the differences in the physiological responses to stress between species or among landraces and cultivars is a premise for planning successful breeding programs for stress-tolerant commercial cultivars [4,5]. Plant roots must exclude most of the Na^+^ and Cl^−^ of the soil solution to avoid salts quickly reach lethal levels in the shoots. Compartmentalization of the retained tiny portion of saline ions in the cell vacuoles and the balancing osmotic pressure of organic solutes in the cytosol assures the required osmotic adjustment to match soil solution [6]. In the short-term, salinity causes osmotic stress in the root-zone, which triggers ABA-mediated stomatal closure that limits photosynthesis [7]. In the long-term, the accumulation of salts in the leaves causes ionic toxicity and ionic imbalances, with disruption of ion homeostasis and oxidative stress [8]. Thus, photosynthesis is further impaired by metabolic limitations [9].

Analysis of the fast fluorescence induction is a reliable and quick non-destructive tool to investigate the photosynthetic apparatus’s structure and function in response to various types of stress [10]. PSII reaction centers (RCs) are open in dark-adapted leaves, as all the stable primary PSII acceptor Q_A_ molecules are oxidized. There is neither water oxidation by PSII nor energy flow in the electron transport chain, and Chl *a* fluorescence emission (F_t_) is at basal level (F_o_). Upon exposure to actinic light, F_t_ shows a polyphasic fast rise (Kautsky effect), with significant steps between the basal level and the peak value, F_P_ (OJIP curve) [11]. During the photochemical phase (O-J), F_t_ rises from F_o_ (point O) to F_J_ after 2 ms (point J), which indicates the single turnover reduction of Q_A_ [12]. During the subsequent much slower thermal phase (J-I), the rise from F_J_ to F_I_ at 30 ms is due to the reduction of the secondary quinone acceptor Q_B_, the PQ pool, and the cytochrome b_6_f complex. At the end of this phase (point I), electrons reach plastocyanin and then ferredoxin at the PSIs electron acceptor side. Their reduction continues in the next slowest thermal phase (I-P) as F_t_ rises to the peak value (F_P_, point P) in less than 1 s. Under saturating illumination, F_P_ represents the maximum emission (F_M_), which indicates the reduction of all Q_A_ molecules and the saturation of electron flow at the acceptor side of PSII. Thus, all RCs are considered closed [11]. Other significant steps can show up in the polyphasic rise, such as the K step at 0.30 ms (point V_K_) (OKJIP curve). Inhibition of the water-splitting system increases V_K_; thus, analysis of Kautsky curves allows investigation of the donor side of PSII [13,14].

In the late seventies, Prof. Reto J. Strasser conceived the theory of energy fluxes in the thylakoid membranes [15] and developed the related Kautsky curve JIP test analysis. The test provides several parameters related to the structure and function of photosystems [12,15], namely PSII and its relationships with PSI functioning [16]. Detailed definitions and equations of raw and derived JIP test fluorescence parameters are reported in Table 1 (see also [11]). Authors have used JIP test to study the effects of various types of stress [17] Faseela at al.2020 [18], such as heat [13,19] drought [20,21,22,23], salinity [24,25], combined drought and salt stress [26,27] ozone [28,29], nutrient deficiency [30,31], toxic metals [10], or climate change [32].

Sweet pepper (*Capsicum annuum*, L.) is an essential horticultural crop in the world as it is in Italy [33]. Pepper grown in either open field or greenhouse is moderately sensitive to salinity [34]. Higher yield and better fruit quality are obtained in glasshouse than open field conditions [35]. In a soilless culture, the response of sweet pepper marketable fruits to the electrical conductivity (EC_w_) of the nutrient solution had a threshold of 2.8 dS m^−1^ and a slope of 7.6% [36]. This species shows some salt tolerance differences among cultivars [37,38,39]. In a long-term salt-stress experiment, [40] Giorio et al. (2020) tested the hypothesis of higher salt stress tolerance of a traditional landrace than a widely spread high-yield commercial cultivar. Nevertheless, the commercial cultivar showed better salt tolerance than the landrace in terms of yield reduction. The accumulation of sodium in the stems was lower in the commercial cultivar than in the landrace. The authors argued that the higher chloride exclusion from leaves permitted more extended photosynthetic apparatus functionality in the commercial cultivar than in the landrace. On this basis and for the same experiment, in this work, we have verified the hypothesis of significant differences between the two sweet pepper genotypes in the structure and function of the photosystems, namely, PSII, assessed through the JIP test analysis of the Chl *a* fluorescence induction transients (OKJIP curves).

Here, we demonstrated that long-term salt stress caused loss of efficiency in the oxygen-evolving complex (OEC) and along the path from light absorption, trapping of excitation energy, and electron transport between PSII and PSI acceptors. These effects occurred earlier and at a greater extent in the landrace than the commercial cultivar. Due to the higher inactivation of PSII reaction centers, the landrace had lower electron transport per sampled leaf area and thus lower CO_2_ fixation capacity than the commercial cultivar. These results corroborate the reported explanation for the difference in the salt-stress tolerance of yield between the two sweet pepper genotypes and confirm the high usefulness of JIP test analysis of polyphasic fluorescence transients to screen for stress tolerance.

## 2. Results and Discussion

Glycophytic species growing in salinized soils struggle with stress conditions along all their life cycle. Therefore, long-term experiments are required to uncover differences in stress tolerance mechanisms between species or cultivars and landraces. Two sweet pepper genotypes, the landrace “Cazzone Giallo” (CG) and the commercial cultivar “Quadrato d’Asti” (QA), were grown with a nutrient solution (NS) (control treatment, S0) or with NS added by 120 mmol liter^−1^ NaCl from 13 days after transplanting (DAT 13) until harvest (DAT 168) (salt treatment, S120). We investigated the function and structure of photosynthetic apparatus through the JIP test analysis of OKJIP curves in response to long-term exposure to salinity.

### 2.1. OKJIP Polyphasic Fluorescence Transients

Stress conditions impair photosynthesis and rouse mechanisms to dissipate excitation energy excess as heat and fluorescence [41]. A tiny portion of the light absorbed by photosystem PSII is re-emitted as Chl *a* fluorescence. However, it is strictly related to the structure and function of photosynthetic apparatus [11]. Double normalization of F_t_ produced the OKJIP curves of V_t_. The V_t_ curves were plotted on a logarithmic time scale to display the physiologically significant steps [42,43] (Figure 1). Assuming no connectivity between PSII units, V_t_ equals the fraction of closed PSII reaction centers ([Q_A_^−^]/[Q_ATotal_]). This ratio drives the kinetics of Chl *a* fluorescence induction, and it is affected directly or indirectly by the rate of all photosynthetic reactions up- and down-stream the oxydoreduction of Q_A_ [11,44,45].

On DAT 82 (Figure 1a), the polyphasic rise of V_t_ was higher in S120 than in control plants, especially in CG, and the differences increased afterward (data not shown). On DAT 167 (Figure 1b), V_K_ (K step) was higher in CG than QA of treatment S120 and higher in S120 than S0 of both genotypes, which implied lower efficiency of the donor side of PSII [14]. Similarly, higher V_J_ (J step) reflected a higher accumulation of reduced primary PSII acceptor Q_A_^−^, thus lower efficiency of the electron transport from Q_A_ to Q_B_ (ψ_ET2o_) [11]. A reduced I-P phase in S120 compared to S0 denoted loss of efficiency of electron flow from primary acceptor Q_A_ until PSI electron acceptors (ψ_RE1o_) [16] (Table 1). Data of F_t_ at the points O, K, J, I and *p* are shown in Appendix A.

### 2.2. Donor Side of PSII

The oxygen-evolving complex (OEC) is one of the most stress-sensitive photosynthetic components [10] and the most sensitive to heat stress [46]. Although not often evident, the K step is considered a natural and always present phenomenon, elicited by any inhibition of the water-splitting system [47]. High salt stress in wheat affected the donor side of PSII more than the acceptor side [48]. OEC inactivation induces an increase in V_K_, a typical feature of heat-stressed leaves [13,47]. K step is often hidden in leaves under weak heat stress or other types of stress. Nonetheless, the increase of V_K_/V_J_, also known as W_K_, revealed OEC inactivation [19]. For instance, it was observed under water stress [20], salt stress [18,25], low-nitrogen fertilization [31], or during leaf senescence [49]. Along the experiment, the increase occurred in S120 plants (Figure 2a), while as expected, V_K_/V_J_ remained proximal to 0.5 in the control plants [19]. Incomplete water splitting with consequent reduction of the electron supply to the reaction centers of PSII leads to an excess of excitation energy and ROS generation, which causes oxidative damage to thylakoid membranes [41,50]. High levels of ROS are quite deleterious to growth and crop yield if there is insufficient scavenging by the enzymatic and the non-enzymatic antioxidants [51]. Drought tolerant cultivars showed more pronounced K step than the less tolerant ones [20]. The authors proposed K step as a potential early stress indicator before the occurrence of any visible symptom. Inhibition of OEC occurred more and earlier in CG than in QA (Figure 2a).

### 2.3. Maximum Quantum Yield, Efficiencies of Electron Transport, and Performance Indexes

After more than two months since starting salinity treatment (DAT 82), the maximum quantum yield of PSII, *φ*_Po_, showed low sensitivity to salt stress (Figure 2b). In S120 of CG, *φ*_Po_ was only 5% lower than the other three G × S combinations, which values were proximal to the optimum 0.83 [52]. It is common to observe little response of *φ*_Po_ to various types of stress as reported for salinity in wheat [48], drought in barley [20], and salinity or drought in sunflower [27]. Since DAT 154, *φ*_Po_ in S120 was statistically significantly lower than S0 in both QA and CG.

The energy-dependent (qE), the state transition (qT), and the zeaxanthin-dependent (qZ) quenching mechanisms dissipate excitation energy excess as heat. They occur within seconds to tens of minutes or up to more than an hour and prevent potential photodamage to photosynthetic apparatus [53]. More precisely, the less the thermal dissipation, the more the photodamage [54]. Under sustained stress conditions, as it was for S120 treatment that showed low CO_2_ assimilation since DAT 102 [40], these mechanisms may result insufficient. Further investigations are required to quantify NPQ mechanisms to confirm this assertion in the two pepper genotypes under our experimental conditions. Excess of excitation energy leads to permanent/slow-reversible reduction of electron acceptors, leakage of electrons, and generation of harmful reactive radicals [41]. To avoid photodamages under high light, plants have also developed photoinhibitory quenching mechanisms (qI). They involve inactivation and degradation of D1 protein in the PSII reaction centers that take hours or longer to turn on and off [53]. The trade-off of these mechanisms is photoinhibition, a light-induced decrease of the quantum yield of photosynthetic CO_2_ assimilation. After DAT 82, *φ*_Po_ decreased in S120 plants of both the two genotypes and at DAT 167 reached the remarkable low values of 0.50 in QA and 0.36 in CG, which indicated a pronounced photoinhibition [52,55]. A similar decrease in *φ*_Po_ measured at midday suggested a marked photoinhibition with associated high energy dissipation in perennial ryegrass under stressful climatic conditions [56]. Alike V_K_/V_J_, salinity affected φ_Po_ earlier and to a greater extent CG than QA (Figure 2b).

Strasser et al. [15,57] introduced the performance index on absorption basis PI_ABS_ that integrates three independent Nernst-type parameters (Table 1), which in chemistry express the ratio between the reduced and the oxidized concentration of a compound [11,24] and references therein. Thus, PI_ABS_ refers to the energy conservation in the path from light energy absorption, excitation energy trapping, and delivery in the intersystem electron transport chain. As γ is the probability for a *Chl* molecule to function as a reaction center, the structure component parameter γ_RC2_/(1 − γ _RC2_) = RC/ABS is the number of active RC per PSII antenna Chl. The component φ_Po_/(1 − φ_Po_) here measured as F_v_/F_o_ refers to the partial performance of primary photochemistry as it represents the ratio between trapped and dissipation energy fluxes TRo/DIo. The component ψ_ET2o_/(1 − ψ_ET2o_) is for the performance of thermal reactions of the intersystem electron carriers as ψ_ET2o_ = ET_2o_/TR_o_ is the efficiency/probability with which an electron trapped in Q_A_ is transferred to Q_B_ (Table 1). To include the photosynthetic performance of electron transport beyond Q_B_, PI_ABS_ was extended to PI_ABS_Total_ [58], which incorporates a fourth component based on δ_RE1o_, the efficiency/probability with which an electron is transferred from Q_B_ to PSI electron acceptors [11].

In contrast to φ_Po_, many papers have reported a rapid decrease of PI_ABS_ in response to various types of stress, such as water deficit [18,20,21,59], salinity [25,48], and a decrease of PI_ABS_Total_ as reported for moderate ozone stress [29]. Even during a daily cycle in a grass species, PI_ABS_ significantly decreased before φ_Po_ on a more than a less stressful day during summer [56]. The two PIs were significantly lower in S120 than S0 in both the two genotypes since DAT 82 and reached at DAT 167 insubstantial tiny values due to the severe stress conditions (Figure 3a,b). Alike both V_K_/V_J_ and φ_Po_, the salinity effect on PIs along the trial was stronger in CG than QA, which further indicated lesser salt stress tolerance of the former than the latter genotype.

The effect of salinity on the PIs components expressed as S120 normalized by S0 (S120/S0) is shown as spider plots in Figure 4. Thus, the lower the normalized value, the higher the salt stress effect in S120, while the normalized parameters equal unit in S0. During the experiment, the decrease of the normalized RC/ABS observed in both genotypes, if not due to an enlargement in the antenna size, can be attributed to the stress-induced conversion of the reaction centers from active (RC) to non-Q_A_ reducing reaction centers, called inactive or silent (RC_silent_) [15]. On DAT 82, the deleterious effect of salt stress on the conservation energy increased (i.e., the normalized parameters decreased), along the path from the trapping of excitation energy in PSII expressed by the normalized component [φ_Po_/(1 − φ_Po_)], to electron transport from Q_A_ to Q_B_ [normalized (ψ*_ET2o_*/(1 − ψ*_ET2o_*)] and from Q_B_ until PSI acceptors [normalized (δ*_RE1o_*/(1 − δ*_RE1o_*)]. Conversely, a reversal order of the salinity effect along the path occurred on DAT 167, which indicated a higher stress sensitivity of PSII than PSI in the long term. These results are in accord with data obtained in plants under Ca-deficiency conditions [30].

Interestingly, conversely to the other three PI-component parameters, the detrimental effect of salinity on δ*_RE1o_*/(1 − δ*_RE1o_*) decreased in the late of the trial (the normalized parameter increased), namely in the cultivar QA. The normalized value on DAT 167 was even higher than on DAT 82. Such a recovery of salt stress effect was not sufficient to significantly alleviate the strong effect on PI_ABStot_. A decrease in δ*_RE1o_* can be due to a reduced I-P phase (1-V_I_) associated to impairment of PSI content or its functioning in the linear electron flow [30]. The recovery was possibly due to the photoprotection effect of the qI mechanisms exerted through the inactivation of PSII reaction centers [53]. Increased photorespiration or increased efficiency of cyclic electron flow around PSI [60,61] may also promote non-photochemical dissipation of energy [62]. Salt stress can reduce stomatal conductance; thus, leaf temperature increases and internal CO_2_ concentration decreases. The latter two effects of salinity enhance RuBisCO oxygenation [63], which irreversibly produce the dangerous hydrogen as reported for the moderately salt-tolerant sugar beet [51]. The protective role of photorespiration is controversial, but it was an important mechanism that avoided photoinhibition in castor bean under salinity [64]. A decrease in PSII electron transport and the increase in non-photochemical quenching associated with alternative electron sinks (i.e., photorespiration and water-water cycle) and cyclic electron flow around PSI was proposed by [62] and recently hypothesized in rice plants under various abiotic stress [18]. However, photorespiration and water-water cycle were negligible at low temperature in grapevine leaves, which showed no photoinactivation of PSII due to a highly efficient thermal dissipation [65]. The authors underlined the necessity to measure the relative contribution of photoprotective mechanisms that avoid/limit inactivation of PSII, namely, ΔpH and xanthophyll-dependent NPQ, photorespiration, water–water cycle (Mehler reactions), D1 repair, and PSI inactivation. Thus, additional investigations would be advisable to further undercover differences in the various photoprotective mechanisms between the two sweet pepper genotypes. The differences between the two genotypes in the salt stress response of the four partial performances along the trial reflected those of PIs as the cultivar QA was less affected than the landrace CG.

### 2.4. Membrane and Leaf Models of Energy Fluxes

The software Biolyzer provided graphical representations of the energy fluxes (pipelines) expressed per active reaction centers RC (specific/membrane model) or per measured area of leaf sample, called excited cross section CS_M_, approximated by F_M_ (phenomenological/leaf model). Exposure to long-term salt stress markedly increased in S120 compared with S0 the specific fluxes of (*i*) the light energy absorption (ABS/RC), (*ii*) the trapped excitation energy (TRo/RC), and (*iii*) the dissipation energy (DIo/RC) (Figure 5).

ABS/RC is a relative measure of the size of antenna that supplies excitation energy to each active reaction center and is here assesses as (M_o_/V_J_)/*φ*_Po_, where M_o_ is the approximate initial slope of V_t_ curve [11]. The pipeline models also indicated that the increase in energy fluxes in S120 plants was associated with the conversion of (active) RC to RC_silent_. Inactivated reaction centers cannot reduce the primary acceptor Q_A_ and do not emit a significant fluorescence amount. They significantly quench fluorescence by competitive heat dissipation of the excitation energy (heat sink centers) [15,45]. RC inactivation also occurred under drought or heat stress [66], salt stress [24,48], and nutrition stress [30]. Inactivation of RC denotes photooxidative damages to protein D1 and/or the manganese cluster of OEC [54]. Compared to the total reaction centers per cross section, the leaf models quantified 22% in QA (Figure 6a) and 45% in CG (Figure 6b) of RC_silent_.

Conversely to ABS/RC, TRo/RC, and DIo/RC, a minor effect of salinity occurred to the specific electron transport ETo/RC (Figure 5). However, the protection mechanism arose at the expense of useful trapping and thus at the expense of energy delivery into the electron transport chain per excited cross section (ETo/CS_M_) (Figure 6). Reduction of ETo/CS_M_ also happened in salt-stressed sweet sorghum [25] and wheat [48], drought- or heat-stressed wheat [66], and senescing flag leaves of rice [49]. Increased heat dissipation and higher number of RC_silent_ resulted in lower electron transport flux per cross section in a less than a more salt-stress tolerant barley landrace [24]. Among six sweet sorghum genotypes, the highest salt-stress tolerant had the highest φ_Po_, PI_ABS_, and ETo/CS_M_ and the lowest V_K_/V_J_ and number of inactive RC [25]. Similarly, we can consider the cultivar QA more tolerant than the landrace CG. The leaf model pipelines indicated that after long-term salinity conditions, ETo/CS_M_ in S120 was halved in QA and reduced to 30% in CG as compared to S0 (Figure 6), so it was the potential capacity of CO_2_ fixation. These data corroborate the reduction in yield under high salinity of 49% in the commercial cultivar and 82% in the landrace reported in a previous paper for the same experiment [40].

### 2.5. Salt Stress Tolerance in Relation with Na^+^ and Cl^−^ Leaf Content

Salt stress can be responsible for reduced photosynthesis, growth, and yield [7,67,68]. Despite the deep knowledge on the salt-stress mechanisms [69], some crucial aspects remain questioned, especially those regarding ion toxicity [70]. In some species, the effects of salt stress were more due to Cl^−^ than Na^+^ as found in fava bean by [71]. The authors reported that impaired photosynthetic capacity was associated with chlorophyll degradation not caused by Na+, disrupting chloride homeostasis in the chloroplasts. We observed a significant stress effect on the chlorophyll content index (CCI) measured on DAT 85 [40]. CCI in the control plants was 13% statistically significantly higher in CG than QA. However, CCI in S120 decreased by 31% in CG as compared to the control, while no stress effect occurred in QA [40]. The green deepness of the leaf model in Figure 6 also shows that long-term salt stress caused a higher reduction in chlorophyll content of CG than QA. Mechanisms to avoid harmful Cl^−^ leaf content can be less efficient than those related to Na^+^. For instance, [68] reviewed that conversely to sodium, chloride re-translocation via phloem from leaves to other organs has not been ascertained. Hence, cytoplasmic homeostasis is limited by the storage capacity of vacuoles. Therefore, leaf Cl^−^ accumulation under long-term salinity eventually results in the loss of cytoplasmic homeostasis. Then chloride-induced inhibition of both dark reaction enzymes and primary photochemistry impairs photosynthetic capacity. Indeed, promoting Cl^−^ exclusion from shoots by substrate drench application of a bioactive molecule induced salt stress tolerance in basil [72]. Therefore, the worse exclusion of chloride from leaves of CG than QA did possibly cause the lower salt stress tolerance despite the better sodium stem compartmentalization [40]. Lines of emmer wheat that exerted two among the three salt tolerance mechanisms of osmotic tolerance, exclusion of Na^+^ from leaves, and its compartmentalization in the vacuoles showed higher salt stress tolerance of growth than those lines that relied only on one mechanism [69]. Moreover, Na^+^ exclusion and its tissue tolerance were mutually exclusive. Further investigations on the two pepper genotypes could focus on whether Na^+^ and Cl^−^ exclusions from leaves compensate for the possibly lower tissue tolerance or the higher tissue tolerance counteracts the lower exclusion. Salt acclimation by a pre-treatment a moderately low saline concentration ameliorated tolerance in maize and wheat, as shown by better water and ionic relations, less sodium accumulation and hydrogen peroxide production, and increased enzymatic antioxidants [73]. Thus, this practice could be advisable for salt-stress sensitive sweet pepper genotypes grown in soilless systems.

## 3. Conclusions

JIP test analysis of OKJIP curves demonstrated the more extended functionality of photosystems under long-term salt stress in the commercial cultivar “Quadrato d’Asti” compared to the landrace “Cazzone Giallo”, which was ascribed to the lower chloride leaf content. Along the trial, the detrimental effects of salinity on the oxygen-evolving complex, the light energy absorption, the trapping of excitons, and the electron transport occurred both earlier and with a greater extent in the landrace than in the commercial cultivar. As expected, the maximum quantum yield of primary PSII photochemistry was less sensitive than the performance indexes. JIP test analysis also indicated double inactivation of PSII reaction centers in the landrace than in the commercial cultivar, with associated higher heat dissipation of excitation energy excess, which limits photodamage. The trade-off of heat dissipation was an impaired energy delivery per sample leaf area and thus impaired potential carbon fixation and yield. A recovery of the electron transport efficiency in the path from the secondary PSII acceptor until PSI acceptors occurred late in the trial. This study can provide valuable information to breeding programs, aiming to select the genes underlying the long-term salt stress tolerance.

## 4. Materials and Methods

### 4.1. Plant Material, Growth Conditions, and Salt Treatments

Two sweet pepper genotypes, the Italian widespread commercial cultivar Quadrato D’Asti (QA) and the landrace Cazzone Giallo (CG) were tested for salt stress tolerance. The experiment was carried out in a greenhouse of CREA-OF (Pontecagnano, SA, Italy) with an automatic computer-controlled soilless system equipped with a drip fertigation device. A basic nutrient solution (NS) was used to grow plants of the control (S0), while the treatment S120 was grown with NS supplemented with 120 mmol NaCl liter^−1^ (treatment S120). The composition of NS was (mN): Na^+^, 0.2; N-NH_4_^+^, 0.5; K^+^, 5.0; Ca^2+^, 10.8; Mg^2+^, 4.0; Cl^−^, 0.5; N-NO_3_^−^, 14.6; P-H_2_PO_4_^−^, 1.2; S-SO4^−^, 3.5; HCO_3_^−^, 0.5. Micronutrients were added according to [74]. The electrical conductivity (EC_w_) of the two nutrient solutions was 2.6 for S0 and 15.6 dS m^−1^ for S120. Addition of nitric acid kept the pH within the 5.5–6.0 range, while addition of fresh water stabilized the EC_w_ of the nutrient solutions. SAIS S.p.A. (Cesena, Italy) and Cooperativa Arca 2010 (Acerra, NA, Italy) provided QA and CG seeds, respectively. Seeds were germinated in a greenhouse at Arca 2010 and transplanted one seedling per pot into a greenhouse 37 days later. Thirty pots per genotype were distributed in the two saline treatments (15 plants per treatment). The salt treatment started 13 days after transplanting (DAT 13) and continued until the yellow fruit stage (DAT 168). The two solutions were pumped each from an independent tank to the corresponding channel and supplied to plants by two emitters per plot (emitter flow rate: 2 L h^−1^). More details on plant growth, NS composition, its supply to and daily consumption by plants, drainage from pots, and recirculation to keep a stable NS concentration in the soilless closed system are reported in [40].

### 4.2. Fast Chlorophyll a Fluorescence Transient and JIP Test

The performances of the structure and function of the photosynthetic apparatus were assessed through the JIP test analysis of the OKJIP fluorescent transients. The measurements were carried out on dark-adapted leaves using a continuous excitation Handy PEA fluorometer (Hansatech Instruments, King-s Lynn, UK). For fluorescence induction, the instrument adopts an excitation light pulse emitted by three (red) 650 nm light diodes, with NIR short pass cut-off filters and a 22 nm spectral-line half width. Light was applied for 1 s at the maximum available PPFD of 3500 μmol (photons) m^−2^ s^−1^ focused on a spot of 5 mm diameter. The fluorescence detector is a high-performance Pin photodiode and associated amplifier circuit. It has a peak wavelength at 650 nm and a spectral line half-width of 22 nm (Hansatech, pers. comm. 13 April 2021) The hardware is capable of detecting fluorescence at a 12-bit resolution. The Ft measurement rate was every 0.01 ms until 0.3 ms from starting of illumination, every 0.1 ms until 3 ms, every ms until 30 ms, every 10 ms until 300 ms, and every 100 ms until 1 s. Leaves were adapted to dark for 30 min using the equipped white leaf clips before assessing the OKJIP curve. Data were downloaded from the instrument to a PC using the Hansatech Pea Plus software ver. 1.01. The F_t_ curve raw data were then exported and processed for JIP test analysis by the software Biolyzer HP3 ver. 3.06, developed and kindly provided to us by Prof. Reto J. Strasser (Bioenergetics Laboratory, University of Geneva, Switzerland). Table 1 reports the main technical data and the JIP test parameters with the related equations. Measurements were taken on two leaves per plant in 12 plant replicates per genotype in the morning (10:00–12:00) of 5 days spaced from DAT 82 to DAT 167.

### 4.3. Statistical Analysis

Kruskal–Wallis test by ranks was carried out on the parameters V_K_/V_J_, *φ*_Po_, PI_ABS_, PI_ABS_Tot_, to evaluate the genotype x salinity (G × S) interaction effects by comparing the mean rank of the four G × S combinations (CG-S0, CG-S120, QA-S0, and QA-S120). When Kruskal–Wallis tests found significant differences, a pairwise Dunn test with Bonferroni corrections was executed using Xlstat 2020 [75].

## Figures and Tables

**Figure 1 plants-10-00887-f001:**
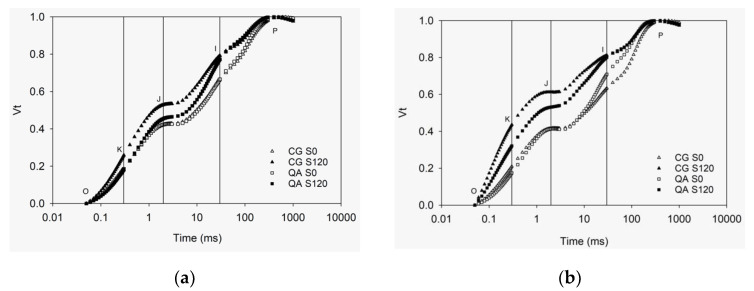
Polyphasic rise of relative variable fluorescence (V_t_) upon illumination with saturating light of dark-adapted leaves of cultivar QA and landrace CG for treatments S0 and S120 measured on DAT 82 (**a**) and DAT167 (**b**). In each panel, the vertical lines intersect the K point at 300 µs (left line), the J point at 2 ms (middle line), and then I point at 30 ms (right line). Values are the means of 24 replicates.

**Figure 2 plants-10-00887-f002:**
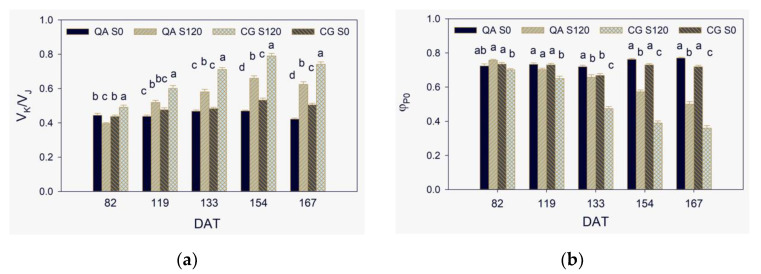
The ratio of V_K_ to V_J_ (**a**) and the maximum quantum yield of primary PSII photochemistry, *φ*_Po_ (**b**) from DAT 82 to DAT 167. Values are the means of 24 replicates. The vertical line on each bar represents the standard error of the mean. Within each DAT, values among the four G × S combinations with the same letter are not significantly different at *p* < 0.05.

**Figure 3 plants-10-00887-f003:**
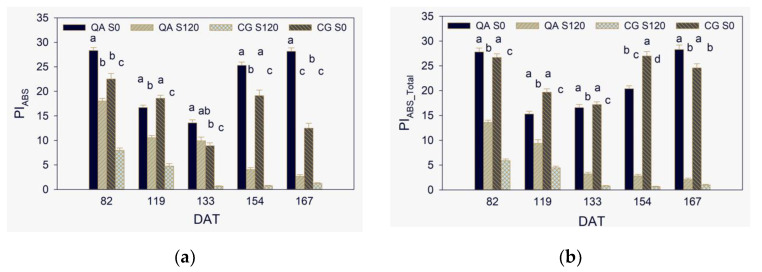
The two performance indexes on absorption basis, PI_ABS_ (**a**) and PI_ABS_total_ (**b**), from DAT 82 to DAT 167. Values are the means of 24 replicates. The vertical line on each bar represents the mean standard error. Within each DAT, values among the four G × S combinations with the same letter are not significantly different at *p* < 0.05.

**Figure 4 plants-10-00887-f004:**
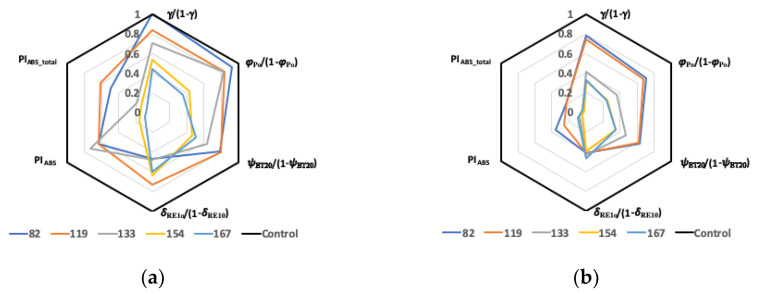
Spider plots of PI_ABS_, PI_ABS_total_, and their components RC/ABS = γ _RC2_/(1 − γ _RC2_), *φ*_Po_/(1 − *φ*_Po_), (ψ*_ET2o_*/(1 − ψ*_ET2o_*) and (δ*_RE1o_*/(1 − δ*_RE1o_*) in QA (**a**) and CG (**b**) from DAT 82 to DAT 167. Values are shown as the ratio S120/S0. The thicker line indicates the relative value (=1) for the control plants (S0).

**Figure 5 plants-10-00887-f005:**
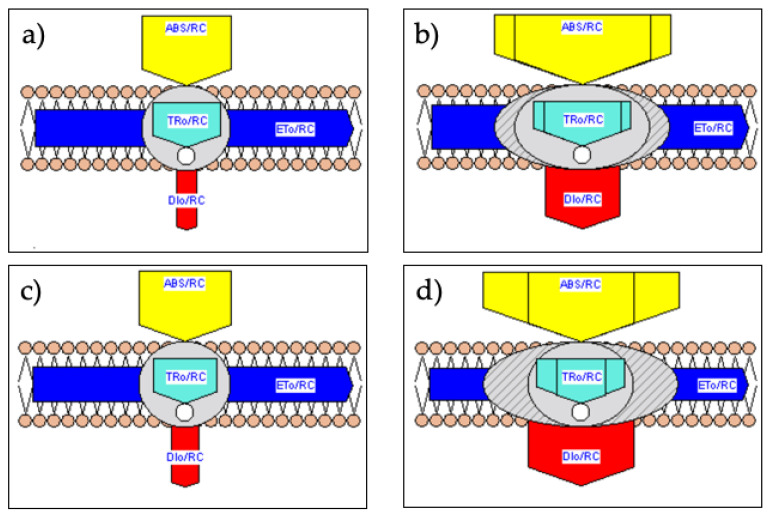
Pipeline models of energy fluxes per active reaction center (membrane/specific model) for S0 (panels **a**, **c**) or S120 (panels **b**, **d**) of QA (panels **a**, **b**) or CG (panels **c**, **d**). Each energy flux is depicted as an arrow, whose width is proportional to the relative flux. The lateral area in the ABS/RC and TRo/CS_M_ represents the energy fluxes related to non-Q_A_ reducing centers (inactive or silent RC). The software Biolyzer produced the pipelines from the average transient curve of 24 replicates measured on DAT 167.

**Figure 6 plants-10-00887-f006:**
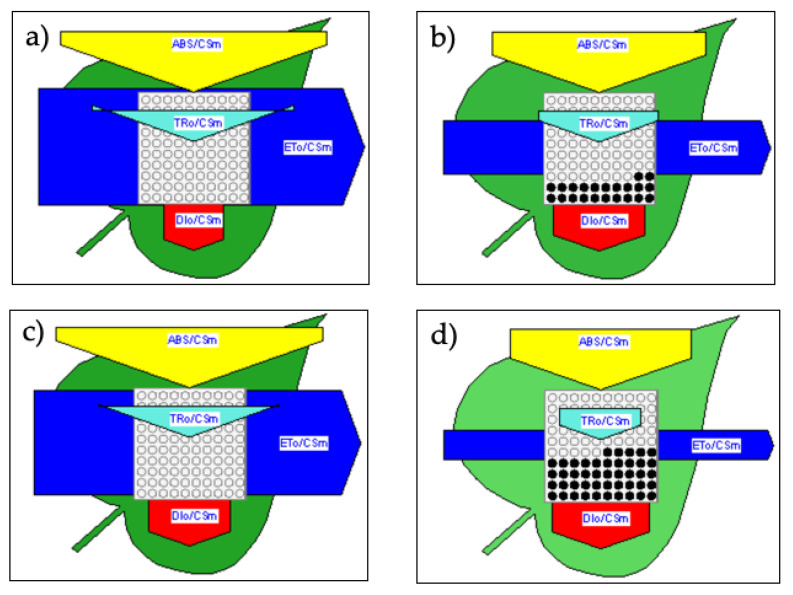
Pipeline models of energy fluxes per excited cross section (CS_M_) of leaf sample, (phenomenological/leaf model) for S0 (panels **a**, **c**) or S120 (panels **b**, **d**) of QA (panels **a**, **b**) or CG (panels **c**, **d**). Each energy flux is depicted as an arrow, whose width is proportional to the relative flux. The deepness of the green color shows chlorophyll concentration per unit leaf surface. The number of active and silent reaction centers is given by the number of white and black circles, respectively. The software Biolyzer produced the pipelines from the average transient curve of 24 replicates measured on DAT 167.

**Table 1 plants-10-00887-t001:** Nomenclature and formulae of the OKJIP transient and related JIP test parameters (modified after [11]).

Raw data obtained from fluorescence induction curve	
**F_t_**	Fluorescence intensity at any time (t) since start of actinic illumination
**F_o_ = F_20μs or 50μs_**	Fluorescence intensity when all PSII RCs are open. The used measure depends on instrumental reliability
**F_300μs_**	Fluorescence intensity at the K step (300 μs) of the induction curve
**F_J_ ≡ F_2ms_**	Fluorescence intensity at the J step (2 ms) of the induction curve
**F_I_ ≡ F_30ms_**	Fluorescence intensity at the I step (30 ms) of the induction curve
**F_P_ ≡ F_M_**	Fluorescence intensity when all PSII RCs are closed due to saturating illumination
Technical fluorescence parameters calculated from the raw data	
**V_v_ = F_t_ − F_o_**	Variable Chl fluorescence
**F_V_ = F_M_ − F_o_**	Maximum variable Chl fluorescence
**V_t_ = (F_t_ − F_o_)/(F_M_ − F_o_)**	Relative variable Chl fluorescence
**M_o_ ≈ (∆V/∆t)_o_ = 4 ms^−1^ (F_300μs_ − F_50μs_)/(F_M_ − F_50μs_)**	Approximate value of the initial slope of **V_t_** curve
Energy fluxes	
**ABS = TR + DI**	Rate of light energy (photons) absorbed by PSII antenna
**TR**	Rate of excitation energy (excitons) trapped by the PSII RCs (causing reduction of the primary PSII acceptor, **Q_A_**)
**TRo**	Maximum TR (initial, at time t = 0)
**DI**	Rate of energy dissipation in the PSIIs, in processes other than trapping
**ET2o**	Electron transport flux from primary (**Q_A_**) to secondary (**Q_B_**) PSII acceptor
**RE1o**	Electron transport flux from **Q_B_** until PSI acceptors (initial, at time t = 0)
Quantum yields and efficiencies/probabilities	
***φ*** **_Po_ ≡ ** **TRo / ABS = 1 − F_o_/F_M_**	Maximum quantum yield of primary PSII photochemistry (initial, at time t = 0)
***φ*** **_ET2o_ ≡ ET2o / ABS = 1 − F_J_/F_M_ = ** ***φ*** **_Po_** **(1 − V_J_)**	Quantum yield of the electron transport flux from **Q_A_** to **Q_B_** (initial, at time t = 0)
***φ*** **_RE1o_ ≡ RE_1o_ / ABS = 1 − F_I_/F_M_ = ** ***φ*** **_Po_** **(1 − V_I_)**	Quantum yield of the electron transport flux until the PSI electron acceptors (initial, at time t = 0)
***ψ*** **_ET2o_ ≡ ET_2o_ / TR_o_ = 1 − V_J_**	Efficiency/probability with which a PSII trapped electron is transferred from **Q_A_** to **Q_B_** (initial, at time t = 0)
***ψ*** **_RE1o_ ≡ RE_1o_ / TR_o_ = 1 – V_I_**	Efficiency/probability with which a PSII trapped electron is transferred until PSI acceptors (initial, at time t = 0)
***δ*** **_RE1o_ ≡ RE_1o_ / ET_2o_ = (1 – V_I_)/(1 – V_J_)**	Efficiency/probability with which an electron from **Q_B_** is transferred until PSI acceptors (initial, at time t = 0)
Specific energy fluxes (per active PSII reaction center)	
**ABS /RC = (M_o_/V_J_) (1/** ***φ*** **_Po_** **)**	Light energy (photons) absorption flux per PSII reaction center (i.e., apparent antenna size of an active PSII)
***γ*** **_RC2_ ≡ ** **Chl_RC_/Chl_Total_**	Probability that a PSII Chl functions as RC
**RC/ABS = ** ***φ*** **_Po_** **V_J_/M_o_ = ** ***γ*** **_RC2_** **/(1 −** ***γ*** **_RC2_** **)**	Number of **Q_A_**-reducing RCs (i.e., active) per PSII antenna Chl
**TR_o_/RC = M_o_/V_J_**	Maximum trapped exciton flux per PSII (initial, at time t = 0)
**ET2o /RC = (M_o_/V_J_) (1 − V_J_)**	Electron transport flux from **Q_A_** to **Q_B_** per PSII RC (initial, at time t = 0)
**RE_1o_ /RC = (M_o_/V_J_) (1 − V_I_)**	Electron transport flux until PSI acceptors per PSII RC (initial, at time t = 0)
Phenomenological energy fluxes/activities (per excited cross section CS)	
**ABS /CS_o_ = F_o_ or ABS /CS_M_ = F_M_**	Light energy (photons) absorption flux per cross section (or also, apparent PSII antenna size)
**RC/CS = (RC/ABS) (ABS /CS)**	The number of active PSII RCs per cross section
**TRo/CS = (TRo/ABS) (ABS /CS)**	Maximum trapped exciton flux per cross section
**ET_2o_/CS = (ET_2o_ / ABS) (ABS /CS)**	Electron transport flux from **Q_A_** to **Q_B_** per cross section
**RE_1o_/CS = (RE_1o_ / ABS) (ABS /CS)**	Electron transport flux until PSI acceptors per cross section
Performance indexes on absorption basis	
**PI_ABS_ = ** **[** ***γ*** **_RC2_** **/(1 −** ***γ*** **_RC2_** **)] [** ***φ*** **_Po_** **/ (1 − ** ***φ*** **_Po_** **)] [** ***ψ*** **_ET2o_ /(1 −** ***ψ*** **_ET2o_)]**	Performance index for energy conservation from photons absorbed by PSII antenna, to the reduction of **Q_B_**
**PI_ABS_Total_ = PI_ABS_ [** ***δ*** **_RE1o_ /(1 −** ***δ*** **_RE1o_)]**	Performance index for energy conservation from photons absorbed by PSII antenna, until the reduction of PSI acceptors

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
