# Peer review of "Polyphasic OKJIP Chlorophyll a Fluorescence Transient in a Landrace and a Commercial Cultivar of Sweet Pepper (Capsicum annuum, L.) under Long-Term Salt Stress"

_plants, 2021, doi:10.3390/plants10050887_

Round 1

Reviewer 1 Report

In figure 2B, the authors should mention that although PSII quantum yield of S120 CG is only slightly lower than in control at 82 DAT it is significantly lower in S120 CG and S120QA after 154 and 167 DAT.

Point 4. Conclusions should be before Point 3. MM. 

Author Response

Dear Reviewer,

Thank you indeed for reviewing the manuscript.

I have uploaded my answers to your valuable comments.

My coauthor and I hope the manuscript amelioration is sufficient for acceptance.

Kind regards.

Pasquale Giorio

Reviewer 2 Report

REVIEW OF THE ARTICLE BY PASQUALE GIORIO AND MOHAMED HOUSSEMEDDINE SELLAMIPOLYPHASIC OKJIP CHLOROPHYLL A FLUORESCENCE TRANSIENT IN A LANDRACE AND A COMMERCIAL CULTIVAR OF SWEET PEPPER (CAPSICUM ANNUUM, L.) UNDER LONG-TERM SALT STRESS” (plants-1180280)

The Authors described response of Capsicum annuum plants to high salt concentrations. They evaluated primary photochemistry of photosynthetic apparatus by the analysis of chlorophyll fluorescence (CF) transient (OJIP curve) registered by a PEA fluorometer. They calculated and compared main parameters of the curves, e.g. maximal photochemical quantum yield of PS II (Fv/Fm parameter), the probability of electron transport beyond QA and beyond PQ, etc. They paid special attention to the additional inflection point K playing the role of additional stress markers in plants. In their article Gioro and Sellami used a routine approach for assessment of photosynthesis by CF measurements, common software and widely used theory of OJIP curve description developed by professor R.J. Strasser, thus, originality of the work is poor. At the same time, new interesting data were demonstrated for C. annuum, therefore the article is a valuable contribution to the field. Introduction, Results and Materials and Methods are satisfied (but can be improved). I have just a serious question about the role of nonphotochemical quenching mechanisms, qE, qT and qI (see specific comments). Statistical analysis is adequate. The article is in scope of the journal. Collectively, I encourage acceptance of the manuscript after revisions.

SPECIFIC COMMENTS

-In Introduction and discussion, the text is generally well-referenced, however it is better to elucidate and compare Your data with OJIP curves parameters from other works of salt stress response, e.g. there are works with Lolium perenne (perennial ryegrass) or Solanum lycopersicon, as well as other plant stress model objects, such as Haematococcus lacustris.

-Please, provide representative primary data for OJIP-cure parameters calculation (ranges and/or median values) as supplementary: Fo, Fm, Fj, Fi.

-Table 1. By definition, the initial slope, Mo, is defined as Mo=(dV/dt)t=0, therefore the symbol ‘=’ should ber replaced to ‘≈’. See e.g. Strasser, 1997 (Your reference [12]).

-l. 91. The statement is wrong. High salt concentrations are actually not a stress for halophylic and halotolerant plants, such as Dunaliella salina or Mesembryanthemum crystallinum. Moreover, low salt concentrations lead to stress in D. salina cells. Please, modulate the sentence.

-l. 104, 1215, 117, 141. The expressions have been given in Table 1, therefore the information such as “Po, here assessed as (FM-Fo)/FM” is excessive.

-Figure 1: Please, indicate points O, K, J, I and P on the graphs.

-l. 105. The information that CF induction curves are in logarithmic time scale is obviously yielded from Figure 1. Moreover, it is well-known. Thus, it is excessive. The same is true about l. 222-224.

-l. 146-160. Regulatory mechanisms of the non-photochemical quenching of the excited chlorophyll states are built up under an actinic light illumination. Therefore, I cannot understand these discussions for dark-adapted cells. You cannot conclude about non-photochemical quenching without analysis of the stationary phase of the CF induction curve. There is no experimental evidence for this in Your data, thus, it is highly speculative. Please, discuss it in more detail. What is about qT mechanism? 

-l. 146-160. Operation time of the qZ mechanism (energy-independent zeaxanthin quenching) may be higher than one hour. 

-l. 146-160. It is better to provide and discuss more common references here, e.g. [35], [39] from Your reference list.

-l. 162. What is “Nerst-type parameters”? Please, explain

-Methods: Please, indicate the duration time of dark adaptation.

-Methods: Please, indicate spectral range of CF detection.

Author Response

(The authors gave the same response as above.)

Reviewer 3 Report

  1. In my point of view, it is necessary to unify terms in the text, for example "JIP test" at lines 14,24,61 and ect, but "JIP-test" at lines 59,66; "K step" , but "J-step", "I-step" at Table 1.
  2. Since the description of abbreviation "DAT" is not given in the M&M section, it is necessary to give the description at the first mention at the beginning of the text (line 96). Moreover, "168 DAT" at line 96 should be replaced with "DAT 168", to unify this data in the text.
  3. The captions in Figure 4 are blurry and very small, it is necessary to improve it for more reader-friendly displaying.
  4. To improve visualization of the results, it would be better to indicate K, J and I points next to the lines in Figure 1.
  5. Even if you link to articles with full descriptions of the methods used [34], you should briefly describe how much the chlorophyll content varied in these sweet pepper genotypes in control and describe NS composition, including pH of solution.
  6. Line 204, Kalaji et al. (2014b) - the format of the reference differs from others in the text, it is better to use [27].
  7. How the appearance of plants (CG and QA genotypes) changed during long-term salinity conditions? Perhaps this information should be indicated in P. 3.1. 

Author Response

(The authors gave the same response as above.)

Reviewer 4 Report

The title and aims of the research are coherent to the scope of the PLANTS journal. The introduction is informative, precise, and comprised of relevant content. The literary structure of the introduction is ok. The philosophy of this work is very good, applicable in applied plant sciences. The manuscript is well structured and balanced and enhances our current understanding.

The abstract contains some general knowledge, however, it must include mostly a key message which authors wish to convey to the audience or a statement which they argue.

The study was well planned and performed and it collects a series of measures on the different growth and physiological parameters.

The experimental methodology was appropriate and scientific, and the analyses were done correctly.  The authors used very progressive techniques and protocols. The data and pictures are understandable.

The authors simply described the potential of chlorophyll fluorescence as a methodological tool and underlined that some chlorophyll fluorescence characteristics can serve as indicators of the flexibility of photosynthetic apparatus under environmental stress.

Some arguments need clearer and tighter presentation. The understanding of mechanisms of salt stress is limited.  Better characterization of the plant physiological reactions under stress conditions, the importance of the adaptability processes, osmotic adjustment, reactions of the photosynthetic apparatus could be useful.

At several places, authors presented fundamental concepts and information with inaccurate scientific explanations.

Authors concluded that the detrimental effects of salinity on the oxygen evolving-complex, the trapping of light energy in PSII and delivering in the electron transport chain, occurred earlier and more in the landrace than the cultivar (probably selected tested cultivar). Do authors generally think that landraces are more sensitive to salt / or other environmental stresses than modern cultivars?

Add more references. In the paper Rastogi et al (2020) we can see a very wide spectrum of genotype differences to salinity. What is a critical level of salinity for sweet pepper plants? I am very surprised with OEC inactivation at 120 mmol.l-1 of salinity.

I would appreciate the analysis of the salt tolerance mechanism. The authors could add new information about crop adaptability and metabolic flexibility, possibilities to use the physiological criteria to improve plant stress tolerance, and useful practical applications for alleviating salt stress in plants.

Specific comments:

- the lack of a clear hypothesis

- standard parameters were determined. I very much appreciate Biolyzer models for the expression of the results.

- line 81 – authors hypothesized that long-term salt stress caused loss of efficiency in the oxygen-evolving complex (OEC). Describe better the reasons, why sweet pepper plants use this adaptation mechanism to salinity stress?

- line 216 – Explain the reasons for increased photorespiration or increased efficiency of cyclic electron flow around PSI

- line 334 – Salt stress-induced the inactivation of RCs in the PSII, and probably also improved both cyclic flow around PSI and photorespiration  – it is speculation. Remove it from the conclusions

- missing critical discussion and a deeper analysis of photosynthetic parameters and processes of osmotic adjustment. The discussion should be improved.

The authors have to discussed more in functioning photosynthetic apparatus in a changing environment. I would like to invite authors to discuss more eco-physiological aspects, and molecular mechanisms using new facts. Please  read/use the following references:

  • Faseela P., Sinisha A.K., et al.: Chlorophyll a fluorescence parameters as indicators of a particular abiotic stress in rice. PHOTOSYNTHETICA 58 (SI): 293-300, 2020, https://doi.org/10.32615/ps.2019.147
  • Zuo Z.Y., Ye F., Wang Z.S., Li S.X., Li H., Guo J.H., Mao H.P., Zhu X.C., Li X.N. (2021): Salt acclimation induced salt tolerance in wild-type and chlorophylb-deficient mutant wheat. Plant Soil Environ., 67: 26–32.
  • Kaya, C.; Ashraf, M.; et al.: The role of endogenous nitric oxide in salicylic acid-induced up-regulation of ascorbate-glutathione cycle involved in salinity tolerance of pepper (Capsicum annuum L.) plants. Plant physiology and biochemistry, 2020, 147, 10-20.
  • Tahjib-Ul-Arif, Abdullah Al Mamun Sohag, Sonya Afrin, et al.: Differential Response of Sugar Beet to Long-Term Mild to Severe Salinity in a Soil–Pot Culture. Agriculture 2019, 9, 223; doi:10.3390/agriculture9100223

Overall, the MS may be accepted for publication in Plants after major revisions as mentioned above.

Author Response

(The authors gave the same response as above.)

Round 2

Reviewer 2 Report

I clearly read responses of the Authors and revised version of the manuscript. I agree with their comments and changes done in the text. Now I feel, that the article is acceptable.

Reviewer 4 Report

I am impressed to see that the authors have taken up all the corrections as suggested and provided necessary clarification. All proposed suggestions were made and the quality and the clarity of the manuscript was substantially improved.

Changes to MS text are adequate, so in my opinion this MS is ready for publication.